# Novel Hydroxyl-Functional Aliphatic CO_2_-Based Polycarbonates: Synthesis and Properties

**DOI:** 10.3390/ijms262010151

**Published:** 2025-10-18

**Authors:** Nikita M. Maximov, Sergey A. Rzhevskiy, Andrey F. Asachenko, Anna V. Plutalova, Elena S. Trofimchuk, Evgenii A. Lysenko, Olga V. Shurupova, Ekaterina S. Tarasova, Elena V. Chernikova, Irina P. Beletskaya

**Affiliations:** 1Faculty of Chemistry, Lomonosov Moscow State University, Lenin Hills, 1, bld. 3, 119991 Moscow, Russia; nmm33@mail.ru (N.M.M.); annaplutalova@gmail.com (A.V.P.); elena_trofimchuk@mail.ru (E.S.T.); evglys1970@gmail.com (E.A.L.); beletska@org.chem.msu.ru (I.P.B.); 2A.V. Topchiev Institute of Petrochemical Synthesis of Russian Academy of Sciences, Leninsky Av., 119991 Moscow, Russia; rs89a@yandex.ru (S.A.R.); shurupovao@yandex.ru (O.V.S.); ekatya99@bk.ru (E.S.T.); 3Faculty of Materials Science, Shenzhen MSU-BIT University, No. 1 International University Park Road, Longgang District, Shenzhen 518172, China

**Keywords:** polycarbonates, ring-opening copolymerization, rac-(salcy)Co^III^X complexes

## Abstract

A series of novel functional polycarbonates, specifically poly(solketal glycidyl ether carbonate-*co*-propylene carbonate)s with varying compositions, were synthesized through the ring-opening copolymerization of solketal glycidyl ether, propylene oxide, and carbon dioxide. The reaction was catalyzed by rac-(salcy)Co^III^X complexes with bis(triphenylphosphine)iminium salts as co-catalysts, achieving high selectivity. The resulting terpolymers exhibited number-average molecular weights ranging from 2 × 10^4^ to 1 × 10^5^ and a narrow, bimodal molecular weight distribution, with dispersities of 1.02–1.07 for each mode. Interestingly, the addition of a small amount of water to the reaction mixture yielded a terpolymer with a unimodal molecular weight distribution and a dispersity of 1.11. Subsequent acidic hydrolysis of the solketal protective groups produced poly(glyceryl glycerol carbonate-*co*-propylene carbonate). All terpolymers were amorphous, with T*_g_* near or below room temperature. The hydroxyl-functional polycarbonates underwent cyclodepolymerization under milder conditions compared to polycarbonates with protected hydroxyl groups.

## 1. Introduction

CO_2_-based polycarbonates have gained significant interest due to their potential for large-scale CO_2_ utilization and the benefits in functionalization modification [1,2,3,4,5,6,7,8,9,10]. These polycarbonates offer advantages such as tunable properties (thanks to a wide range of accessible epoxides) [11,12,13], biodegradability [14,15,16], (cyclo)depolymerization capability [17,18,19], and open- or closed-loop recyclability [20]. Among the various CO_2_-based polycarbonates, functional polycarbonates containing OH-groups are of particular interest [21,22]. Their free hydroxyl groups enable functionalization with different chemotherapeutic agents, antibacterial compounds, anti-inflammatory agents, fluorescent tags, and more [23,24].

However, the direct synthesis of polycarbonates with side OH-groups from the monomers seems impossible [25,26,27,28]. Consequently, indirect synthetic routes are necessary. [29,30]. An effective alternative is the ring-opening copolymerization (ROCP) of glycidyl ethers with CO_2_, followed by deprotection. Glycidyl ethers, which are readily synthesized from epichlorohydrin and the corresponding alcohols, are key monomers for this approach [21,31,32,33,34,35,36,37,38,39,40,41,42,43,44,45,46,47,48,49,50].

Zinc catalysts are commonly employed for the ROCP of glycidyl ethers with CO_2_ to yield polymers with high carbonate linkage contents [13,33]. However, these catalysts typically produce polycarbonates with a broad molecular weight distribution (MWD) and Đ_M_ > 2, with the number-average molecular weight (*M_n_*) generally not exceeding 3 × 10^4^. Conversely, Co salen complexes have emerged as effective alternatives, demonstrating enhanced selectivity for polycarbonates over cyclic carbonates, increased molecular weights, and narrower MWDs with Đ_M_ < 1.4 [13,32]. The limited application of cobalt salen complexes stems from a misconception that cobalt catalysts are unsuitable for this reaction, as the coordination of both glycidyl ether oxygens to the metal center leads to catalyst deactivation [48].

To avoid backbone degradation, poly(glycidyl ether carbonate)s must be deprotected under mild conditions to introduce hydrophilic –OH groups [21,32,34,36,48]. The two principal methods for this transformation are catalytic hydrogenation with Pd/C [32,48] and acidic hydrolysis using an ion-exchange resin [34,36] (Figure 1).

The deprotection of poly(glycidyl ether carbonate)s with different carbonate units can significantly affect their mechanical and physicochemical properties across a broad range. Frey et al. [48] reported the synthesis of a series of amorphous poly(glycidyl methyl ether carbonate-*co*-benzyl glycidyl ether carbonate) terpolymers using a ZnEt_2_/pyragallol catalyst. However, this process suffered from low selectivity, generating substantial cyclic carbonates as byproducts. Following deprotection, the resulting poly(glycidyl methyl ether carbonate-*co*-glycerol carbonate)s swelled in water but remained insoluble, despite their high hydroxyl group content. These OH-functionalized polycarbonates with *M_n_* = 9.8 × 10^3^ and Đ_M_ = 3.3, containing approximately 30 mol.% of glycerol carbonate units, exhibited low stability, degrading in the solid state under moist conditions and in DMF solution at room temperature within 10 weeks. In contrast, atactic poly(1,2-glycerol carbonate), derived from poly(ethoxy ethyl glycidyl ether carbonate) with *M_n_* = 17 × 10^3^ and Đ_M_ = 1.46 or from poly(benzyl glycidyl ether carbonate) with *M_n_* = 13.7 × 10^3^ and Đ_M_ = 1.11, degraded more rapidly, with a half-life t_1/2_ of approximately 2–3 days [32,36]. Frey et al. [34] also described terpolymers derived from solketal glycidyl ether, glycidyl methyl ether, and CO_2_. When catalyzed by a Zn-based system, these polycarbonates exhibited *M_n_* = (12–15) × 10^3^, Đ_M_ = 2.5–3.3, a high content of carbonate linkages, and a SolGE molar fraction between 33 and 66%. Notably, unlike other OH-functionalized polycarbonates, poly(glycidyl methyl ether carbonate-*co*-glycidyl glycerol carbonate) containing 33% of glycidyl glycerol carbonate units retained stability in THF solution for up to 3 weeks. This enhanced stability was attributed to the reduced stability of the cyclic carbonate.

The terpolymerization of propylene oxide (PO), 2-[[(2-nitrophenyl)methoxy]-methyl]oxirane, and CO_2_ catalyzed by zinc glutarate, followed by deprotection through UV irradiation, produced poly(propylene carbonate-*co*-glycerol carbonate) of varying compositions [46]. The resulting OH-functionalized terpolymers (containing 1–10 mol % glycerol carbonate units) showed higher T_g_ relative to their protected counterparts, attributed to H-bonding with ether oxygen atoms (or the oxygen atom of the ether group). Additionally, these polymers decreased the water contact angle from 74° to 61°.

Poly(glycidyl ether carbonate)s themselves are attractive materials irrespective of their protective groups, particularly as polymer electrolytes for solid-state lithium-ion batteries [37,38,39]. For instance, poly(glycidyl ether carbonate)s with phenyl, *n*-butyl, *tert*-butyl, ethyl, isopropyl, octyl, stearyl, and methoxyethyl side groups, containing 10 mol.% of LiTFSI, show conductivity comparable to polyether-based electrolytes such as poly[oligo(oxyethylene glycol) methacrylate]/LiTFSI [37,38,39]. Additionally, poly(glycidyl ether carbonate)s also demonstrate valuable adhesive and self-healing properties [43,49].

The full potential of OH-functionalized polycarbonates and their precursors poly(glycidyl ether carbonate)s remains unexplored despite their appealing properties. In this study, we suggest a new route to the synthesis of a series of poly(propylene carbonate-*co*-glycerol carbonate) through the terpolymerization of solketal glycidyl ether (SolGE), racemic PO, and CO_2_ followed by deprotection of OH-groups by acidic hydrolysis. Basing on our previous study [51], a salen Co(III) complex as a catalyst and bis(triphenylphosphine)iminium salt as a co-catalyst were used to conduct the terpolymerization. We suppose that these terpolymers should possess high adhesive properties and may be valuable as polymer electrolytes for solid-state lithium-ion batteries. However, in the current research, we focused our efforts on the controlled synthesis of the terpolymers, their characterization, and evaluation of their recyclability through cyclodepolymerization before and after deprotection.

## 2. Results and Discussion

### 2.1. Terpolymerization of Solketal Glycidyl Ether, Propylene Oxide, and Carbon Dioxide

Functional polycarbonates with OH side groups, such as poly(1,2-glycerol carbonate) and poly(glyceryl glycerol carbonate), cannot be synthesized directly from their corresponding epoxides and CO_2_ [31], necessitating a protecting group strategy. Herein, poly(glyceryl glycerol carbonate-*co*-propylene carbonate) was synthesized through the terpolymerization of SolGE, PO, and CO_2_, followed by the deprotection of solketal glycidyl carbonate. The *rac*-salcyCo^III^X complex and [PPN]Y (X = DNP¯ (**1a**), CF_3_C(O)O¯ (**1b**); Y = Cl¯ (**2a**), CF_3_C(O)O¯ (**2b**)) were used as the catalyst (Cat) and co-catalyst (co-Cat), respectively (Figure 1). The Cat/co-Cat system was selected based on our previous work, where it demonstrated high selectivity for carbonate linkages and a high polymer-to-cyclic carbonate ratio in the ring-opening copolymerization of *rac*-PO and CO_2_ [51].

The molar fraction of SolGE in the monomer feed was varied from 0.1 to 0.9 to control the number of functional groups in the resulting copolymers. All terpolymerizations were carried out under identical solvent-free conditions at room temperature, with ([PO] + [SolGE]):[1a]:[2a] = 4500:1:1, and at 2.5 MPa CO_2_ pressure. The terpolymers were characterized by ^1^H NMR spectroscopy (Figure 2). Signals at δ = 5.00 and 4.0–4.4 ppm correspond to –CH– (b) and –CH_2_– (a) backbone protons, respectively. Isopropylidene protecting group signals (g) and the methyl group (h) of poly(propylene carbonate) appeared at δ = 1.32–1.38 ppm. Protons associated with the –CH_2_–O– group (c, d) appeared at δ = 3.51–3.68 ppm. Finally, the protons of the solketal group CH_2_–CH–O– (e) and –CH–CH_2_–O– (f) were observed at δ = 3.5–4.4 ppm. Importantly, the absence of signals in the region of ~3.40–3.45 ppm, characteristic of polyether backbone protons, confirms the high selectivity for carbonate units’ formation.

Additional verification by ^13^C NMR spectroscopy confirmed the formation of polycarbonate, indicated by the characteristic carbonate resonance at δ = 154 ppm (Appendix A). SolGE is consumed faster than PO during the terpolymerization with CO_2_, irrespective of the monomer feed composition. This trend is consistent at low, moderate, and high conversions (Table 1 and Table 2). The correlation between monomer feed and copolymer composition for all systems is presented in Appendix A.

The reactivity ratios were determined using both the non-linear least squares (NLLS) method [52] and the Fineman–Ross (FR) method [53]: *r*_SolGE_ = 1.8 ± 1.0 and *r*_PO_ = 0.64 ± 0.32 (NLLS); *r*_SolGE_ = 2.7 ± 1.3 and *r*_PO_ = 0.88 ± 0.25 (FR). Both methods indicate that SolGE is more reactive than PO in the terpolymerization (*r*_SolGE_ > 1, *r*_PO_ < 1). An increase in the SolGE content of the monomer feed was found to reduce the selectivity for polycarbonate over cyclic carbonate at initial monomer conversions (Table 1). However, this effect diminished at higher monomer conversions, where the selectivity for both epoxides exceeded 90% (Table 2). The observed turnover frequency (TOF) is consistent for cobalt salen-catalyzed copolymerization of PO [54] or functionalized ethylene oxides and CO_2_ [32], and it decreases slightly with increasing monomer conversion.

All synthesized terpolymers exhibited a bimodal MWD, independent of copolymer composition (Figure 3a). The separation of the two peaks in the MWD (Figure 3b, Table 2) allows for estimation of the average molecular weights and dispersities of the different propagating chain types. Both the low- and high-molecular-weight fractions displayed narrow MWDs. Similar results have been observed earlier for PO copolymerization with CO_2_ mediated by a similar catalyst–co-catalyst system [51,54,55].

A detailed kinetic study of the reaction mixture with *f*_Sol_ = 0.32 at room temperature confirmed the higher reactivity of SolGE over PO, as evidenced by its faster consumption (Figure 4a). The system rapidly reached steady-state conditions, and the polymerization kinetics adhered to a first-order rate law for both monomers (lines 1 and 2), which is characteristic of a living polymerization system (Figure 4b). Consequently, the increased activity of SolGE compared to PO leads to a higher rate of SolGE incorporation into the copolymer. The observed decrease in the SolGE mole fraction in the copolymer with increasing conversion—approaching 0.32 at high conversion—is consistent with the trend calculated from the reactivity ratios of SolGE and PO (Figure 4c).

The MWD of the polycarbonates remains bimodal throughout the terpolymerization, with both modes consistently shifting towards higher molecular weights (Figure 5). The formation of a polymer with bi- or polymodal MWD in catalytic polymerization, a special case of which is the ring-opening copolymerization of epoxides and CO_2_ under the action of organometallic catalysts, is usually due to the coexistence of several types of active centers that differ in their reactivity [56]. Recently, we have shown that neither racemic (salen)CoX and epoxide nor ligand exchange or polymerization conditions (temperature, CO_2_ pressure, or solvent) were responsible for bimodality of MWD [55]. As shown in Figure 6a, the *M_n_* increases linearly with monomer conversion for both the low- and high-molecular-weight fractions, which is characteristic of a living polymerization mechanism ($M_{n}\sim \;conversion\cdot {\left[M\right]}_{0}/{\left[\mathrm{Cat}\right]}_{0}$) and accords with kinetic findings (Figure 4b). The slopes of the *M_n_*–conversion dependence differ by approximately 1.7 between the two fractions, which is close to that discovered previously for the copolymerization of PO and CO_2_ under similar conditions [51]. Furthermore, the dispersity Đ_M_ is low for the low-molecular-weight fraction (1.03–1.05), while it increases from 1.09 to 1.28 for the high-molecular- weight fraction (Figure 6b). The generally recognized mechanism of this process involves the coordination of an epoxide onto a cobalt atom, followed by the activation of the epoxide by a catalyst or co-catalyst anion and the formation of an alkoxide. Then, CO_2_ is incorporated and a carbonate is formed. Repeating these reactions leads to the increase in the M_n_ of a polymer during polymerization and the formation of a polymer with a narrow MWD.

Various reactivities of epoxides in terpolymerization with CO_2_ could provoke the development of compositional inhomogeneity of copolymers. However, in the case of the living mechanism of the process, the compositional inhomogeneity between macromolecules is replaced by the formation of a gradient copolymer (compositional inhomogeneity is developed within macromolecule, from its head to its tail) similarly to living anionic polymerization and controlled radical copolymerization [57,58,59]. We also suppose that both low- and high-molecular-weight fractions have similar compositions, i.e., are compositionally homogeneous. The reasons for this assumption are as follows. The ring-opening copolymerization of PO and CO_2_ exhibits similar features of MWD transformation with the increase in monomer conversion [55], as was described above. The same results we observed for SolGE copolymerization with CO_2_. Hence, both active centers are able to produce polycarbonates, and both epoxides are able to participate in the formation of both fractions. Fractionation of these copolymers by MWs and analysis of the composition of both fractions could confirm this assumption. However, fractionation in this case seems impossible due to very narrow molecular weight distribution (Table 2).

The observed bimodal MWD is characteristic of the ROCP of PO/CO_2_, initiated by cobalt–salen complexes with or without PPN salts [56,60,61,62]. This behavior originates from the existence of two distinct active sites, resulting from the meridional and facial coordination of the salen ligand to the metal center (two conformers, Figure 2) [63]. As we have proposed earlier, the catalytic activities of complexes with different conformations differed, resulting in the bimodal MWD of polycarbonate formed [55]. Thus, one can obtain a polycarbonate with a unimodal MWD by suppressing the formation of one of the conformers. Indeed, as we have shown previously, the equilibrium between these two conformers can be shifted towards the formation of a facial conformer through the addition of a ppm amount of water or alcohols [55]. The additive serves as a coordinating ligand, fixes the facial conformation (Figure 2), and converts the bimodal MWD into a unimodal one.

In the case of PO/CO_2_ copolymerization, the optimal molar ratio [H_2_O]/[catalyst] providing the formation of the polycarbonate with a unimodal MWD was equal to 20 [55]. Thus, we chose this ratio in the terpolymerization of SolGE/PO/CO_2_. Figure 7 shows the MWD of the terpolymers synthesized with *f*_SolGE_ = 0.11 without (1) and with (2) H_2_O, using **1b**/**2b** as a Cat/co-Cat. A twenty-fold excess of H_2_O relative to the catalyst yields a terpolymer with a unimodal MWD, albeit with a lower MW (Table 3). Consistent with the ROCP of PO and CO_2_, the addition of water does not affect the reaction selectivity and copolymer composition. The latter result confirms that monomer reactivity with respect to both sites is similar, and we may propose that the terpolymers formed are compositionally homogeneous.

### 2.2. Deprotection of Poly(solketal glycidyl ether carbonate-co-propylene carbonate)

The solketal protecting group was removed to release the two hydroxyl groups in the polycarbonate macromolecule, forming glyceryl glycerol carbonate units (Figure 3).

Prior to deprotection, each polymer sample was purified by triple precipitation from chloroform into methanol to remove residual catalyst and prevent rapid depolymerization. The deprotection was performed using an acidic ion-exchange resin in a THF/methanol mixture. The extent of deprotection (ED) was monitored by NMR spectroscopy and calculated using Equation (1).(1)$$ED=\left(1-\frac{I2-6I1+I3}{2I3-6I1}\right)*100\%$$

Figure 8 and Figure 9 present the ^1^H and ^13^C NMR spectra, respectively, of the copolymers with a molar content of SolGE carbonate units of *f*_SolGE_ = 0.11 before (1) and after (2) deprotection. The disappearance of the signals corresponding to the solketal protecting group’s >C(CH_3_)_2_ moiety—specifically, the proton signals at 1.38 ppm (Figure 8), and at 25.49, 26.81, and 109.49 ppm (Figure 9)—verifies complete deprotection.

The extent of deprotection increased with reaction time, reaching completion after approximately 24 h (Table 4). The reaction proceeded under relatively mild conditions, with no associated decrease in molecular weight (Appendix A, Table 4). The stability of the deprotected polycarbonate, however, was found to be composition-dependent. For example, sample 3 (Table 4) maintained its molecular weight characteristics even after one month in THF, both before and after deprotection.

In contrast, samples 11, 13, and 14 (Table 4) completely degraded in THF within 24 h at room temperature. However, these polymers remained stable in DMSO-*d_6_* solution. The absence of degradation products was confirmed by the ^1^H NMR spectrum (Figure 10), which showed no cyclic carbonate signals.

### 2.3. Properties of Poly(solketal glycidyl ether carbonate-co-propylene carbonate) Before and After Deprotection

All poly(solketal glycidyl ether carbonate-*co*-propylene carbonate) samples are readily soluble in chloroform, CH_2_Cl_2_, acetone, and DMF. Dissolution in THF and DMSO, however, requires up to 24 h. These polymers remain insoluble in methanol, water, and diethyl ether. Deprotected samples containing ~10 mol.% of glyceryl glycerol carbonate units retain solubility in these solvents, albeit with slower dissolution. In contrast, poly(glyceryl glycerol carbonate-*co*-propylene carbonate) with ~53% of glyceryl glycerol carbonate units is insoluble in chloroalkanes but swells in water and methanol.

The thermal properties of polycarbonates were investigated using DSC and TGA techniques. The synthesized terpolymers were amorphous and exhibited T*_g_* at moderate values. Representative DSC thermograms for poly(solketal glycidyl ether carbonate-*co*-propylene carbonate)s are shown in Figure 11. The T*_g_* was found to decrease with both an increasing molar content of SolGE units in the terpolymer (Figure 11a, Table 4) and a decreasing molecular weight of the terpolymers (samples 4–6, Table 4). A similar trend of decreasing *T_g_* with a higher SolGE content was observed in [34] for polycarbonates derived from the SolGE/glycidyl methyl ether/CO_2_ terpolymerization.

The observed reduction in *T_g_* is likely attributable to an increase in free volume resulting from the bulky steric solketal group. An additional decrease in *T_g_* occurs due to the deprotection (Figure 11c, Table 5). At elevated temperatures (above 220 °C), endothermic effects associated with polycarbonate degradation are observed. The temperature of the maximum endothermic effect increases with a higher SolGE content (Figure 11b, Table 5). An increase in SolGE content correlates with a rise in the temperature at which the maximum endothermic effect occurs (Figure 11b, Table 5). A new peak also emerges, the intensity of which grows with an increasing SolGE content. This secondary peak diminishes after the deprotection.

This first endothermic effect is likely associated with chain degradation and cyclodepolymerization, while the second corresponds to the breakdown of cyclic carbonates. The deprotected polycarbonate exhibits lower thermal stability than its precursor (Figure 11c). When heated in air instead of argon, the thermal behavior of poly(solketal glycidyl ether carbonate-co-propylene carbonate)s changes (Appendix A). Under these oxidative conditions, weight loss begins in a similar temperature range irrespective of polymer composition, likely due to new oxidative reactions occurring in the air. To compare the thermal stability of the protected and deprotected polycarbonates, an additional experiment was performed. The polycarbonates were heated at 180 °C under vacuum for a predetermined time and subsequently analyzed by the SEC (Figure 12) and NMR spectroscopy (Appendix A). Both the high-molecular-weight (Figure 12a, curve 2) and the low-molecular-weight (Figure 12b, curve 2) terpolymers (*f*_SolGE_ = 0.11) remained stable after heating for 1 h at 180 °C, with the absence of detectable cyclic carbonate signals in the NMR spectra (Appendix A).

Extended heating for 3 h released small amounts of propylene carbonate (6.8 mol.%) and glycidyl glycerol carbonate (7.9 mol.%), while the MWD of the terpolymer remained nearly unchanged (Figure 12b, curve 3). In contrast, the deprotected terpolymer exhibited significantly reduced thermal stability. After just 1 h of heating, its molecular weight decreased by a factor of three (Figure 12b, curve 4), and the product contained 48 mol.% of propylene carbonate and 81 mol.% of glycidyl glycerol carbonate.

## 3. Materials and Methods

### 3.1. Materials

All air- or water-sensitive reactions were conducted under dry nitrogen using standard Schlenk-line techniques or in a drybox. Methylene chloride, *n*-hexane (“Component-reaktiv”, Moscow, Russia; “dry”), and diethyl ether (“Component-reaktiv”, Moscow, Russia; “reagent grade”) were dried and degassed by passing through a column of activated alumina followed by sparging with dry nitrogen. *Rac*-propylene oxide, PO (“Sigma-Aldrich”, St. Louis, MO, USA; for synthesis), was dried over calcium hydride, distilled under argon, and transferred under vacuum prior to use. Epichlorohydrin (“Sigma-Aldrich”, St. Louis, MO, USA; >99.0%), tetrabutylammonium bromide (“Sigma-Aldrich”, St. Louis, MO, USA; >99.0%), and 1,2-isopropylidene-*rac*-glycerol (solketal, “Sigma-Aldrich”, St. Louis, MO, USA; >97.0%) were used as received. The catalyst *rac*-(salcy)Co^III^DNP (salcy = N,N-bis(3,5-di-*tert*-butylsalicylidene)-1,2-diaminocyclohexane, DNP = 2,4-dinitrophenoxy), *rac*-(salcy)Co^III^O(O)CCF_3_, and the co-catalysts, [PPN]Cl and [PPN]O(O)CCF_3_ ([PPN] = bis(triphenylphosphine)iminium), were prepared according to literature procedures [51].

### 3.2. Synthesis of Solketal Glycidyl Ether

2,2-Dimethyl-4-((oxiran-2-ylmethoxy)methyl)-1,3-dioxolane (solketal glycidyl ether, SolGE) was synthesized using a modified procedure (Figure 4) [64]. A mixture of solketal (90 g, 0.68 mol) and tetrabutylammonium bromide (12 g) was dissolved in *n*-hexane (450 mL) and combined with a 50% aqueous solution of NaOH (90 mL). Epichlorohydrin (126 g, 1.37 mmol) was dissolved in n-hexane (90 mL) and added to the reaction mixture. The resulting solution was vigorously stirred at 80 °C for 12 h, then diluted with water (200 mL). The target product was extracted with ethyl acetate (2 × 300 mL). The combined organic layers were dried over anhydrous Na_2_SO_4_ and concentrated under reduced pressure. The yield was 83 g or 65%.

^1^H NMR (400 MHz, Chloroform-*d*) δ 4.28 (h, *J* = 5.9 Hz, 1H), 4.10–4.00 (m, 1H), 3.85–3.77 (m, 1H), 3.76–3.69 (m, 1H), 3.67–3.48 (m, 2H), 3.48–3.37 (m, 1H), 3.15 (dd, *J* = 5.8, 2.9 Hz, 1H), 2.79 (t, *J* = 4.6 Hz, 1H), 2.64–2.54 (m, 1H), 1.42 (s, 3H), 1.36 (s, 3H).

^13^C[48] NMR (101 MHz, Chloroform-*d*) δ 109.6, 74.8, 74.7, 72.6, 72.4, 72.3, 66.7, 50.9, 50.8, 44.2, 26.8, 25.5.

Prior to polymerization, the product was purified by vacuum distillation over CaH_2_.

### 3.3. Polymer Synthesis

Typical procedure for terpolymerization.

A mixture of *rac*-(salcy)Co^III^DNP (0.006 mmol) and [PPN]Cl (0.006 mmol) was dissolved in a *rac*-PO/SolGE mixture (14.3 mmol/15.9 mmol) in a 10 mL vial equipped with a magnetic stir bar. After stirring to form a homogeneous red–brown solution, the vial was then placed in a pre-dried 50 mL autoclave. The reactor was pressurized to 2.5 MPa CO_2_, and the mixture was stirred for 24 h at room temperature. The reaction was quenched by depressurizing the autoclave. The resulting mixture was dissolved in 10 mL of CH_2_Cl_2_, and a 50 μL aliquot was removed for conversion and selectivity analysis by ^1^H NMR spectroscopy. The main volume was then precipitated from a CH_2_Cl_2_/MeOH (10/1, *v*/*v*) mixture into MeOH, and this process was repeated three times. The resulting polymer was dried under vacuum to a constant weight. Other reaction compositions are provided in Appendix A (ESI).

Conversion: 79.5% PO, 82% SolGE. Yield: 2.6 g (72%).

^1^H NMR (400 MHz, Chloroform-*d*) δ 5.00 (s, 1H), 4.53–3.91 (m, 3.14H), 3.77–3.41 (m, 2.83H), 1.38–1.17 (m, 4.70H).

^13^C NMR (101 MHz, Chloroform-*d*) δ 154.3, 109.5, 74.6, 74.6, 74.3, 72.6, 72.5, 69.3, 69.1, 66.6, 66.1, 65.8, 26.8, 25.5, 16.3.

The procedure for calculating the conversion and selectivity of the polymerization process from the NMR data (Appendix A) is given in ESI.

Polymerization procedure using an additive:

A mixture of *rac*-(salcy)CoIII O(O)CCF_3_ (0.047 mmol) and [PPN]O(O)CCF_3_ (0.047 mmol) was dissolved in a mixture of *rac*-propylene oxide (204.9 mmol), solketal glycidyl ether (SolGE, 22.65 mmol), dichloromethane (14 mL), and toluene (14 mL) in a 100 mL beaker equipped with a magnetic stir bar. After stirring until a homogeneous red–brown solution formed, an additive (H_2_O, 1 mmol) was introduced. The vial was then placed into a pre-dried 200 mL autoclave, pressurized to 2.5 MPa CO_2_, and stirred for 100 h at room temperature. The reaction was quenched by depressurizing the autoclave. The resulting mixture was dissolved in 100 mL of CH_2_Cl_2_, and a 50 μL aliquot was removed for conversion and selectivity analysis by ^1^H NMR spectroscopy. The main volume was then precipitated from a CH_2_Cl_2_/MeOH (10/1, *v*/*v*) mixture into MeOH, and this process was repeated three times. The resulting polymer was dried under vacuum to a constant weight.

Conversion: 74.1% PO, 97.0% SolGE. Yield: 19.3 g (74%).

^1^H NMR (400 MHz, Chloroform-d) δ 5.00 (s, 1H), 4.43–3.99 (m, 2.29H), 3.77–3.41 (m, 0.58H), 1.38–1.17 (m, 3.41H).

### 3.4. Deprotection of Copolymers Containing SolGE Carbonate Units

The solketal protecting groups were removed by adding 10 wt% acidic ion exchange resin (Dowex 50WX8) to a 10 wt% polymer solution in MeOH/THF (50/50 *v*/*v*) (Figure 2). The mixture was stirred at 40 °C for 24 h. The resin was then removed by filtering the solution through a 0.45 μm PTFE filter, and the polymer was lyophilized under vacuum. The degree of deprotection was assessed using ^1^H NMR spectroscopy.

### 3.5. Depolymerization

Thermal degradation of copolymers containing protected and deprotected SolGE carbonate units was carried out at 180 °C. The copolymer (50 mg) was placed into an ampoule, degassed under vacuum, and sealed. The sealed ampoule was heated in a preheated sand bath at 180 °C for the desired duration. After cooling to room temperature, the ampoule was unsealed. The residue was analyzed by size exclusion chromatography (SEC) and NMR spectroscopy.

### 3.6. Methods

NMR spectra were acquired on a Bruker Avance III HD (400 MHz ^1^H, 101 MHz ^13^C). Chemical shifts are reported relative to the residual undeuterated solvent peaks. Coupling constants *J* are given in Hertz as positive values regardless of their actual sign. Signal multiplicities are abbreviated as follows: s (singlet), d (doublet), t (triplet), m (multiplet), and br (broadened).

The SEC measurements were performed in THF at 40 °C with a flow rate of 1.0 mL/min using a 1260 Infinity II GPC/SEC Multidetector System chromatograph (Agilent, Santa Clara, CA, USA) equipped with two PLgel 5 μm MIXED B columns. The SEC system was calibrated with narrow-dispersity linear poly(methyl methacrylate) standards with MW ranging from 800 to 2 × 10^6^.

Differential scanning calorimetry (DSC) measurements were carried out on a DSC 200L (XiangYi Instruments, Xiangtan, Hunan, China) using 40 μL Al pans. The polymer samples (8–15 mg) were heated under argon at 10 °C/min. Prior to measurement, the samples were preheated to +100 °C, followed by cooling to –50 °C at a constant rate of 10 °C/min. Thermogravimetric analysis was conducted on a Mettler TA4000 system under air at a heating rate of 10 °C min^−1^.

## 4. Conclusions

Novel hydroxyl-functional aliphatic polycarbonate terpolymers with varying compositions were successfully synthesized through the living ROCP of solketal glycidyl ether, *rac*-propylene oxide, and carbon dioxide, followed by deprotection of the hydroxyl group. The combination of catalyst and co-catalyst, comprising *rac*-(salcy)Co^III^DNP and [PPN]Cl, ensured high selectivity, producing polymers without ether groups in the backbone and a cyclic carbonate molar fraction of less than 10%. Kinetic studies confirmed the higher reactivity of SolGE compared to propylene oxide (*r*_SolGE_ > 1 and *r*_PO_ < 1) and revealed first-order kinetics in both monomers.

All terpolymers displayed a narrow, bimodal MWD regardless of the copolymer composition. *M_n_* increased linearly with monomer conversion for both low- and high-molecular-weight fractions, which, along with polymerization kinetics, aligns with a living polymerization mechanism. The addition of a catalytic amount of a coordinating additive, such as H_2_O, converted the bimodal MWD into a unimodal distribution with low dispersity (Đ_M_ < 1.1).

The acidic hydrolysis of the solketal protecting group successfully yielded the target polycarbonates with diol side chains. Both the protected and deprotected polycarbonates were amorphous, with a low *T_g_* ranging from –3 to +38 °C. While the protected polycarbonates remained stable in solution, the stability of the deprotected ones varied depending on their composition and solvent. Specifically, terpolymers with a low fraction (~0.1) of hydroxyl-functionalized carbonate units remained stable in THF for over a month, whereas those with a higher fraction (>0.4) degraded in the same solvent. The functionalized polycarbonates degrade under milder conditions compared to polycarbonates with protected hydroxyl groups, enhancing their potential in various applications. Additionally, these polycarbonates can be modified by reacting their hydroxyl groups with aldehyde or ketone groups of functional molecules. The subsequent acid catalyzed the release of these functional molecules, which positions these terpolymers as promising candidates for various biomedical applications.

## Data Availability

The data supporting this article have been included as part of the Appendix A.

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
