# Peer review of "Novel Hydroxyl-Functional Aliphatic CO2-Based Polycarbonates: Synthesis and Properties"

_ijms, 2025, doi:10.3390/ijms262010151_

Round 1
Reviewer 1 Report
Comments and Suggestions for Authors
The work of Asachenko, Chernikova and co-authors explores the catalytic ring-opening copolymerization of of solketal glycidyl ether, propylene oxide, and carbon dioxide by salen type cobalt complexes with iminium salts as co-catalysts acidic hydrolysis of the obtained copolymers produced amorphous poly(glyceryl glycerol carbonate-co-propylene carbonate). Additionally, all terpolymers were depolymerized under mild conditions.
The article is well-written, showcasing logical reasoning, rigorous execution, and good quality standards. The conclusions drawn are firmly rooted in the presented data.
According to the reviewer the paper can be accepted in the present form after very few minor corrections, according to the following comments.
- Scheme 1: add square brackets to the last structure as well.
- Figure 4: add letters on the peaks to make them easier to assign.
- In tables 1 and 2 specify the temperature at which the reactions were carried out.
- Besides the experiment reported in Table 3, which involved adding a twenty-fold excess of water compared to the catalyst, were other experiments conducted with different amounts of water?
In these cases, what masses were measured? These experiments could provide insights into how the presence of water affects these reactions.
Author Response
Rev. 1
1) Scheme 1: add square brackets to the last structure as well.
Reply: The brackets were added.
2) Figure 4: add letters on the peaks to make them easier to assign.
Reply: corrected
3) In tables 1 and 2 specify the temperature at which the reactions were carried out.
Reply: corrected
4) Besides the experiment reported in Table 3, which involved adding a twenty-fold excess of water compared to the catalyst, were other experiments conducted with different amounts of water? In these cases, what masses were measured? These experiments could provide insights into how the presence of water affects these reactions.
Reply: Previously, we conducted a study on the copolymerization of propylene oxide and CO2 in the presence of salen cobalt complexes. The salen complexes used were (rac)-(salcy)CoX (salcy = N,N´-bis(3,5-di-tert-butylsalicylidene)-1,2-diaminocyclohexane) and (salen)CoX (salen = N,N´-bis(3,5-di-tert-butylsalicylidene)ethylenediamine), where X = OC(=O)CF3, 2,4-dinitrophenolate (DNP) and co-catalysts were bis(triphenylphosphine)iminium salts, [PPN]Y, where Y = COOCF3, Cl, DNP (Ref.57 in the current paper). We found that a twenty-fold molar excess of a coordinating additive, H2O, relative to the catalyst was optimal for producing a unimodal molecular weight distribution of poly(propylene carbonate). Therefore, we used this ratio in our current work, and we added a reference to this information and explained the choice of [H2O]/[catalyst] ratio in the text.
Reviewer 2 Report
Comments and Suggestions for Authors
Review of the article “Novel hydroxyl-functional aliphatic CO2-based polycarbonates: synthesis and properties” by Irina P. Beletskaya etc.
The article by Irina P. Beletskaya and co-authors is devoted to a very important and relevant topic of using CO2 to produce polymers that can be used in various fields of materials science: biomedical chemistry, in electrolytes for ion polymer batteries, etc. The use of modern catalytic systems capable of polymerizing monomers under mild conditions, especially with the incorporation of CO2, is extremely interesting. The team of authors has carried out a lot of work on the synthesis and analysis of the data obtained using advanced methods. There are no questions about the methodology and use of catalytic systems. Most of the misunderstandings and questions arose when reading the part on polymerization.
Comments:
- The introduction section is too long and contains too many details of previous works from the literature. At the same time, the problem statement and the relevance of this work, on the contrary, are not described in detail. It is necessary to remove unnecessary details of other works and bring the relevance in more detail.
- On Figure 4. The signals themselves are not signed, although a description of the structure is given. It should be noted that the “a” signals shown in the red and blue parts are neither chemically nor magnetically equivalent.
- The abbreviation PO is not deciphered anywhere.
- Table 2 is not sufficiently discussed in the text.
- The presence of a bimodal distribution indicates the presence of two polymerization processes and/or the secondary formation of nonlinear structures. Why didn't the authors analyze polymer fractions with higher and lower MM? How was the compositional uniformity of the copolymers proved? Have polymer products been fractionated and fractions analyzed?
- How did the authors prove the absence of branches in polymer products?
- The arguments presented in the text, which relate to the copolymerization processes of SolGE and PO, are not obvious. From calculations using the Fineman-Ross method, in the case when r1 > 1, r2 < 1, polymer enrichment with monomer units 1 should be observed first, and, when it is exhausted, with units 2. From the above discussions, it is unclear what happens in the case of SolGE and PO? In the case of fSolGE = 0.32, according to kinetic curves, does the conversion of both monomers increase? is there a growth of two homopolymers at different rates? A more detailed explanation is needed in the text.
- How was the conversion of monomers measured? The text does not mention this.
- P9L248 the authors write "The addition of a catalytic amount of a coordinating additive, such as H2O or an alcohol, converts the bimodal MWD into a unimodal one". Has the polymer composition been studied in this case? If not, why not?
Conclusion
The work of the team of authors leaves a contradictory feeling. On the one hand, it is clear that the authors have done a lot of work on the relevant topic. Interesting and promising catalytic systems have been used for the synthesis of copolymers. On the other hand, the evidence component of the polymer part raises a number of questions, which raises more questions than answers. If the terpolymers are heterogeneous in their composition, as indicated by the bimodal distribution, and have not been individually purified in any way, all further studies (TGA, DSC) do not make sense. And it is not correct to talk about their merits. The article can be published only in the case of convincing and evidence-based answers on the polymer part or when conducting additional experiments on fractionation and purification, and then characterization by DSC methods (major revision).
Author Response
Rev.2
- The introduction section is too long and contains too many details of previous works from the literature. At the same time, the problem statement and the relevance of this work, on the contrary, are not described in detail. It is necessary to remove unnecessary details of other works and bring the relevance in more detail.
Reply: We have shortened the text in the introduction and have emphasized the novelty of the research
- On Figure 4. The signals themselves are not signed, although a description of the structure is given. It should be noted that the “a” signals shown in the red and blue parts are neither chemically nor magnetically equivalent.
Reply: corrected
- The abbreviation PO is not deciphered anywhere.
Reply: corrected
- Table 2 is not sufficiently discussed in the text.
Reply: corrected
- The presence of a bimodal distribution indicates the presence of two polymerization processes and/or the secondary formation of nonlinear structures. Why didn't the authors analyze polymer fractions with higher and lower MM? How was the compositional uniformity of the copolymers proved? Have polymer products been fractionated and fractions analyzed?
Reply: Polymerization mediated by organometallic complexes typically proceeds via a living mechanism, which is confirmed by linear increase of Mn vs monomer conversion. In some cases, the catalytic polymerization can lead to the formation of bi- and even polymodal MWD of the polymer due to the presence of several types of active centers. The typical example is the Ziegler-Natta catalysts. In contrast to olefin polymerization mediated by the Ziegler-Natta catalysts, the transfer to polymer resulting in the formation of branches is impossible in the case of ring-opening copolymerization of epoxides and CO2 (the corresponding discussion and references are added to the text). In the studied system, MWs of both high- and low-molecular weight fractions increases with monomer conversion confirming 1) the living mechanism of the process; 2) the presence of 2 types of growing active centers. Concerning the compositional homogeneity of both fractions, we may suppose that they have similar composition. The reasons for this assumption are as follows. 1) The ring-opening copolymerization of PO and CO2 exhibits the similar features of MWD transformation with the increase of the monomer conversion and the high selectivity of carbonate unit’s formation is observed. The same results were observed for SolGE copolymerization with CO2. Hence, both active centers are able to produce polycarbonates. 2) The living nature of the process should provide the formation of gradient copolymer (rather than compositional inhomogeneous) similarly to living anionic polymerization and controlled radical or controlled cationic copolymerization. Thus, the composition inhomogeneity between macromolecules is replaced with composition inhomogeneity within macromolecule, from its head to its tail. Fractionation of these copolymers by MWs seems impossible due to narrow molecular weight distribution (see Table 2 in the paper). Fractionation of the copolymers by composition seems also impossible due to the solubility/insolubility of poly(propylene carbonate) and poly(solketal glycidyl ether carbonate) in the same solvents. A corresponding discussion on polymerization mechanism, the nature of bimodality and composition homogeneity of copolymers has been added to the text of the article.
- How did the authors prove the absence of branches in polymer products?
Reply: All the copolymers were investigated by 1H and 13C NMR spectroscopy. The typical 13C NMR spectrum of the synthesized poly(solketal glycidyl ether carbonate-co-propylene carbonate) from the monomer feed containing fSolGE = 0.53 is given in Figure S1 (ESI). We have not detected the signals corresponding to the quaternary carbon atom, which is in accordance with the mechanism of ring-opening copolymerization of epoxides and CO2.
- The arguments presented in the text, which relate to the copolymerization processes of SolGE and PO, are not obvious. From calculations using the Fineman-Ross method, in the case when r1 > 1, r2 < 1, polymer enrichment with monomer units 1 should be observed first, and, when it is exhausted, with units 2. From the above discussions, it is unclear what happens in the case of SolGE and PO? In the case of fSolGE = 0.32, according to kinetic curves, does the conversion of both monomers increase? is there a growth of two homopolymers at different rates? A more detailed explanation is needed in the text.
Reply: The two homopolymers can grow in the case when both r1 > 1 and r2 > 1. In the case of r1 > 1 and r2 < 1 the more active monomer is consumed faster than the less active one. In the case of the living polymerization it results in the formation of gradient copolymer. This conclusion is in accordance with one glass transition temperature observed by DSC. The discussion is added to the text.
- How was the conversion of monomers measured? The text does not mention this.
Reply: The description is given in the ESI file. We have added the corresponding reference to ESI in the text.
- P9L248 the authors write "The addition of a catalytic amount of a coordinating additive, such as H2O or an alcohol, converts the bimodal MWD into a unimodal one". Has the polymer composition been studied in this case? If not, why not?
Reply: The copolymer composition was evaluated from 1H NMR spectra of the polymers. The corresponding column was added to the Table 3.
We have carefully revised the paper taking into account all the questions and comments of the reviewer.

Reviewer 3 Report
Comments and Suggestions for Authors
The manuscript by Maximov et al., “Novel Hydroxyl-Functional Aliphatic COâ‚‚-Based Polycarbonates: Synthesis and Properties,” reports the synthesis of new hydroxyl-functional aliphatic polycarbonates via the ring-opening terpolymerization of solketal glycidyl ether, propylene oxide, and COâ‚‚ catalyzed by Co-salen complexes. This study aims to advance sustainable COâ‚‚ utilization through the design of functional polycarbonates with tunable molecular weight and thermal behavior. The work is timely, well-organized, and supported by comprehensive structural and thermal characterization. However, several aspects of the experimental methodology, data interpretation, and mechanistic conclusions would benefit from clarification and additional evidence to fully substantiate the claims. The following points are offered as constructive feedback.
Major Comments
- Reproducibility and Experimental Details- The study would be more robust if the polymerization procedures included details on reproducibility and batch-to-batch consistency. It is recommended to report replicate runs or provide statistical error margins for yields, conversions, and selectivity values.
- Evidence for Scheme 5 and Reaction Control - Additional experimental or literature evidence supporting the proposed mechanism in Scheme 5 would improve confidence in the interpretation. Furthermore, clarification on how critical operational parameters—particularly temperature control and COâ‚‚ pressure regulation—were maintained would be valuable, as these strongly influence catalyst activity and selectivity.
- Polymerization “Living” Character - The claim of a “living polymerization” mechanism is insufficiently supported. The observed bimodal molecular weight distributions suggest multiple active sites or chain-transfer processes rather than a single controlled propagation. Including quantitative Mn versus conversion data and dispersity evolution plots would help verify kinetic control. End-group or MALDI-TOF analyses are strongly recommended to provide direct mechanistic evidence of living behavior.
- SEC Calibration and Data Interpretation - Discuss the limitations of SEC calibration using PMMA standards, as polycarbonates typically have smaller hydrodynamic volumes. This likely leads to an overestimation of Mn values and should be addressed or corrected where possible.
- Degradation and Stability Studies - The evidence for degradation and stability differences between protected and deprotected polymers is mostly qualitative. Incorporating time-dependent molecular weight data or kinetic degradation profiles would substantiate the observed behavior. Additionally, clarifying the degradation pathway (e.g., cyclodepolymerization versus backbone cleavage) and explaining solvent-dependent stability trends (THF vs. DMSO) would improve mechanistic understanding.
- Context and Application Relevance - Benchmarking the reported materials against comparable Zn- and Co-based catalytic systems would better highlight the novelty and performance advantages. The discussion on biomedical relevance is intriguing but currently speculative; a brief demonstration of post-functionalization, solubility behavior, or biocompatibility testing would strengthen the practical impact of the study.
Author Response
Rev. 3.
- Reproducibility and Experimental Details- The study would be more robust if the polymerization procedures included details on reproducibility and batch-to-batch consistency. It is recommended to report replicate runs or provide statistical error margins for yields, conversions, and selectivity values.
Reply: Reproducibility of the experiments was demonstrated for the copolymerization of PO/SolGe monomers in 7:3 ratio in several experiments over 24 h and 28 h (including kinetic experiments). However, slightly different monomer conversions and degrees of polymerization were observed at the same reaction time, which is due to similar, but still different, monomer/catalyst ratios (The determining factor here is the error in weighing milligram quantities of catalyst, ∆=0.2 mg per 5 or 10 mg). Moreover, regardless of the degree of polymerization, the samples were characterized by very similar compositions. These data indicate that the polymer composition is reproducible from experiment to experiment and depends primarily on the initial monomer ratio. Furthermore, this is confirmed by a combination of data from other experiments with varying monomer contents. The accuracy of determining polymer composition, selectivity, and conversion is defined by the error of the NMR method.
Quantitative 1H-NMR (qHNMR) is a powerful tool for analysis of natural products (10.1021/np200993k), pharmacological substances (10.1039/c9ay01403a) and polymers (10.1007/s00216-018-1510-z 10.1039/D0AN00879F).
All NMR experiments were performed on a Bruker Avance III HD (400 MHz) spectrometer at 298K. 1H NMR spectra were acquired at frequency 400.13 MHz using 30 degrees pulse, 20 ppm spectral window, O1P 10 ppm, 4.09 s acquisition time, 10 s relaxation delay, 1 dummy scan and 4 scans. For each spectrum, automatic baseline correction and automatic phase correction were performed. For analysis, peaks belonging to the polymer (5.0 ppm) and two monomers (3.0 and 3.15 ppm) that did not overlap with other peaks were selected. Integrals of equal width and at least 75 Hz were used. Signal-to-noise ratio were determined using “sino” command in the Topspin 3.5 or using script “SNR Peak Calculator” in the MNova 11.2.
Typically, the signal-to-noise ratio is the primary source of error when all other criteria are met (10.1021/jm500734a) (e.g., full spin recovery between scans) for qHNMR. Moreover, a good signal-to-noise ratio can help achieve minimal uncertainties for quantitative NMR (10.1016/j.jmr.2018.11.004). We used SNR to estimate the relative and absolute uncertainty between two peaks:
(ΔR/R)2 = (ΔA/A)2 + (ΔB/B)2 = (1/SNRA)2 + (1/SNRB)2
In this case, the smaller of the two SNRs calculated for each peak was taken for the estimate and the relative uncertainty was calculated relative to the peak at 3.15 ppm.
4.98 ppm – SNR 312, 3.14 ppm – SNR 1089, 1.34 ppm – SNR 17614.
Thus, for SNRA = 312, SNRB = 1089 we obtain the relative uncertainty ΔR/R = 0.33%, and absolute uncertainty is: ΔR=0.33*(SNRB/SNRA)=±1.16%. For methyl groups peak the relative uncertainty is 0.09%.
Assuming both peaks are integrated with roughly the same width, the uncertainty in two integral ratio roughly matches that for the peak height ratio estimate above.
- Evidence for Scheme 5 and Reaction Control - Additional experimental or literature evidence supporting the proposed mechanism in Scheme 5 would improve confidence in the interpretation. Furthermore, clarification on how critical operational parameters—particularly temperature control and COâ‚‚ pressure regulation—were maintained would be valuable, as these strongly influence catalyst activity and selectivity.
Reply: We have significantly expanded the part devoted to the polymerization and provided it with more details.
- Polymerization “Living” Character - The claim of a “living polymerization” mechanism is insufficiently supported. The observed bimodal molecular weight distributions suggest multiple active sites or chain-transfer processes rather than a single controlled propagation. Including quantitative Mn versus conversion data and dispersity evolution plots would help verify kinetic control. End-group or MALDI-TOF analyses are strongly recommended to provide direct mechanistic evidence of living behavior.
Reply: We have separated the peaks (see Fig. 3b as an example) and have plotted Mn and dispersity versus conversion for each mode (Fig. 6). The linear dependence of Mn vs conversion and formation of polymers with narrow MWD is considered an unambiguous proof of a living polymerization mechanism (please see Controlled and Living Polymerizations. From Mechanisms to Applications. Ed. by A.H.E. Müller and K. Matyaszewski, Wiley-VCH, 2009). Concerning end-group analysis, it is impossible to detect end-groups for high-molecular weight polymers. The similar restriction has MALDI-TOF analysis.
- SEC Calibration and Data Interpretation - Discuss the limitations of SEC calibration using PMMA standards, as polycarbonates typically have smaller hydrodynamic volumes. This likely leads to an overestimation of Mn values and should be addressed or corrected where possible.
Reply: We agree with the reviewer that the determined molecular weight characteristics are relative values. The calibration curve is linear in the all range of the studied MWs. The average MWs were determined using refractive index detector. There is no information about the values of M-K-H parameters for poly(propylene carbonate) in THF. Thus, it seems impossible to recalculate MWs from PMMS calibration. However, we would like to address attention of the reviewer to the fact that presentation of relative MWs is usual and widely used in majority of publications.
- Degradation and Stability Studies - The evidence for degradation and stability differences between protected and deprotected polymers is mostly qualitative. Incorporating time-dependent molecular weight data or kinetic degradation profiles would substantiate the observed behavior. Additionally, clarifying the degradation pathway (e.g., cyclodepolymerization versus backbone cleavage) and explaining solvent-dependent stability trends (THF vs. DMSO) would improve mechanistic understanding.
Reply: We agree with the reviewer that these experiments are useful. However, we prefer to present the kinetics and mechanism of thermal destruction of the synthesized terpolymers in another paper. This mechanism is more complicated, and random scission of macromolecules (random destruction) occurs firstly followed by cyclodepolymerization as we have observed for poly(propylene carbonate).
- Context and Application Relevance - Benchmarking the reported materials against comparable Zn- and Co-based catalytic systems would better highlight the novelty and performance advantages. The discussion on biomedical relevance is intriguing but currently speculative; a brief demonstration of post-functionalization, solubility behavior, or biocompatibility testing would strengthen the practical impact of the study.
Reply: We agree with the reviewer that this demonstration would strengthen the practical impact of the study. However, in the current research we prefer to focus on the controlled synthesis of the terpolymers, their characterization and evaluation of their recyclability through thermal destruction. Instead of presenting a demonstration at this stage, we prefer to carefully test the terpolymers in various applications and present the results later.
Round 2
Reviewer 2 Report
Comments and Suggestions for Authors
The authors made the recommended corrections and answered the questions sufficiently. I recommend the article to be accepted in IJMS